# Evolved bacterial resistance to the chemotherapy gemcitabine modulates its efficacy in co-cultured cancer cells

**Serkan Sayin[1], Brittany Rosener[1], Carmen G Li[1], Bao Ho[1], Olga Ponomarova[1], Doyle V Ward[2,3], Albertha JM Walhout[1,4], Amir Mitchell[1,3,4,5]***

[1]Department of Systems Biology, University of Massachusetts Chan Medical School, Worcester, United States; [2]Department of Microbiology and Physiological Systems, University of Massachusetts Chan Medical School, Worcester, United States; [3]Program in Microbiome Dynamics, University of Massachusetts Chan Medical School, Worcester, United States; [4]Program in Molecular Medicine, University of Massachusetts Chan Medical School, Worcester, United States; [5]Department of Molecular, Cell and Cancer Biology, University of Massachusetts Chan Medical School, Worcester, United States

**Abstract** Drug metabolism by the microbiome can influence anticancer treatment success. We previously suggested that chemotherapies with antimicrobial activity can select for adaptations in bacterial drug metabolism that can inadvertently influence the host's chemoresistance. We demonstrated that evolved resistance against fluoropyrimidine chemotherapy lowered its efficacy in worms feeding on drug-evolved bacteria (Rosener et al., 2020). Here, we examine a model system that captures local interactions that can occur in the tumor microenvironment. Gammaproteobacteria-colonizing pancreatic tumors can degrade the nucleoside-analog chemotherapy gemcitabine and, in doing so, can increase the tumor's chemoresistance. Using a genetic screen in *Escherichia coli,* we mapped all loss-of-function mutations conferring gemcitabine resistance. Surprisingly, we infer that one third of top resistance mutations increase or decrease bacterial drug breakdown and therefore can either lower or raise the gemcitabine load in the local environment. Experiments in three *E. coli* strains revealed that evolved adaptation converged to inactivation of the nucleoside permease NupC, an adaptation that increased the drug burden on co-cultured cancer cells. The two studies provide complementary insights on the potential impact of microbiome adaptation to chemotherapy by showing that bacteria–drug interactions can have local and systemic influence on drug activity.

*For correspondence: amir.mitchell@umassmed.edu

Competing interest: The authors declare that no competing interests exist.

## Editor's evaluation

This fundamental work advances our understanding of how bacteria evolve to resist drugs used for cancer treatment and how this could potentially affect drug efficacy and treatment outcome. The data were collected and analyzed using a solid methodology and can be used as a starting point for functional studies of the interaction between the microbiome and cancer drug treatment. The findings will be of broad interest to microbiologists and organismal biologists interested in the role of microbiomes in drug responses.

## Introduction

Clinical research on the influence of intratumor bacterial infection can be dated back to more than 150 years ago (*Sepich-Poore et al., 2021*). However, in the past decade, research of the

tumor-microbiome gained significant momentum with the maturation of DNA sequencing technologies and advancement of microbiome research. Multiple recent studies of bacterial colonization in human tumors outlined the magnitude of this phenomenon (reviewed in *Sepich-Poore et al., 2021*; *Goodman and Gardner, 2018*; *Cullin et al., 2021*) Collectively, these works establish that the proportion of infected tumors greatly varies across tumor types and that many tumors harbor microbiomes with a distinctive and characteristic composition of bacterial species. In some cases, as in breast and pancreatic cancer, more than 60% of tumors harbored a tumor-microbiome (*Nejman et al., 2020*). The microbiome, in turn, is known to influence cancer disease through multiple independent mechanisms, including promotion of neoplastic processes in healthy host cells, modulation of the host antitumor immune response, and by bacterial biotransformation of anticancer drugs (*Cullin et al., 2021*; *Roy and Trinchieri, 2017*; *Alexander et al., 2017*; *Riquelme et al., 2019*).

Bacterial metabolism of xenobiotics, including breakdown of host-targeted drugs, is prevalent (*Spanogiannopoulos et al., 2016*). Estimates from recent drug screens show that two thirds of human-targeted drugs can be metabolized by at least one bacterial species that is present in the human gut microbiome (*Zimmermann et al., 2019*). Yet, these interactions are reciprocal, bacteria both metabolize the host-targeting drugs and are also frequently impacted by them (*Zimmermann et al., 2021*). Roughly 25% of host-targeted drugs are potent inhibitors of bacterial growth at physiological concentrations (*Maier et al., 2018*). This proportion is doubled for antineoplastic drugs and almost all anticancer drugs that belong to the antimetabolite drug class have potent antimicrobial activity (*Maier et al., 2018*). A key underexplored question that arises from these reciprocal drug–microbiome interactions is how they impact one another given the ability of microorganisms to evolve and change over short time scales within the host (*Zhao et al., 2019*; *Lieberman, 2022*; *Garud et al., 2019*; *Snitkin et al., 2013*; *Gatt and Margalit, 2021*). Specifically, given that bacteria rapidly evolve resistance to antimicrobial drugs, it is plausible that adaptation to host-targeting drugs that are also antimicrobial will alter bacterial drug metabolism or its transport (*Rosener et al., 2020*; *Kyono et al., 2022*). Such adaptions have been repeatedly observed with standard antibiotics (*Alekshun and Levy, 2007*). In such cases, evolved resistance in tumor-colonizing bacteria may increase or decrease drug availability to the tumor cells, which, in turn, may interfere with the efficacy of the chemotherapy.

The tumor-microbiome in pancreatic cancer has attracted much attention recently due to the prevalence of infections in pancreatic ductal adenocarcinoma (PDAC) (*Nejman et al., 2020*; *Riquelme et al., 2019*; *McAllister et al., 2019*; *Geller et al., 2017*; *Aykut et al., 2019*; *Pushalkar et al., 2018*). Studies have uncovered multiple independent mechanisms through which microbes influence oncogenesis (*Aykut et al., 2019*; *Pushalkar et al., 2018*), disease progression (*Riquelme et al., 2019*), and treatment success (*Geller et al., 2017*) in the pancreas. Bacterial infection is attributed to retrograde bacterial migration from the gastrointestinal tract into the pancreas (*McAllister et al., 2019*; *Pushalkar et al., 2018*). Characterization of the PDAC tumor-microbiome by 16S rRNA gene sequencing showed that proteobacteria are highly enriched relative to the gut microbiome and that they are highly prevalent in pancreatic tumors (*Nejman et al., 2020*; *Geller et al., 2017*; *Pushalkar et al., 2018*). Recent work suggested that pancreatic colonization can impede therapy with gemcitabine (2',2'-difluoro 2' deoxycytidine [dFdC]), a frontline chemotherapy drug that is used for PDAC treatment (*Geller et al., 2017*). Further clinical data provided circumstantial evidence indicating that this interaction may indeed take place in treated patients (*Meriggi and Zaniboni, 2021*; *Mohindroo et al., 2021*; *Gao et al., 2020*).

Gemcitabine drug metabolism is well-understood in the model gamma-proteobacteria *Escherichia coli* (*Geller and Straussman, 2018*; *Figure 1A*). The antimetabolite gemcitabine, a nucleoside analog, is imported into the bacterial cell through the nucleoside transporter NupC and is then phosphorylated. Gemcitabine triphosphate may be incorporated into a newly synthesized DNA strand and then may interfere with chain elongation by masked chain termination (similar to mammalian cells; *de Sousa Cavalcante and Monteiro, 2014*). Therefore, despite its clinical use as an anticancer drug, gemcitabine's mechanism of action potentially makes it a broadly toxic, antimicrobial compound. Previous works showed that some bacterial species can rapidly convert gemcitabine into the less toxic metabolite 2',2'-difluoro-2'-deoxyuridine (dFdU) (*Geller et al., 2017*; *Vande Voorde et al., 2014*; *Lehouritis et al., 2015*). In gamma-proteobacteria, gemcitabine degradation proceeds through a specific isoform of the cytidine deaminase enzyme (Cdd$_L$) (*Geller et al., 2017*). The well-characterized interactions between tumor cells, gemcitabine, and gamma-proteobacteria puts forth a good model

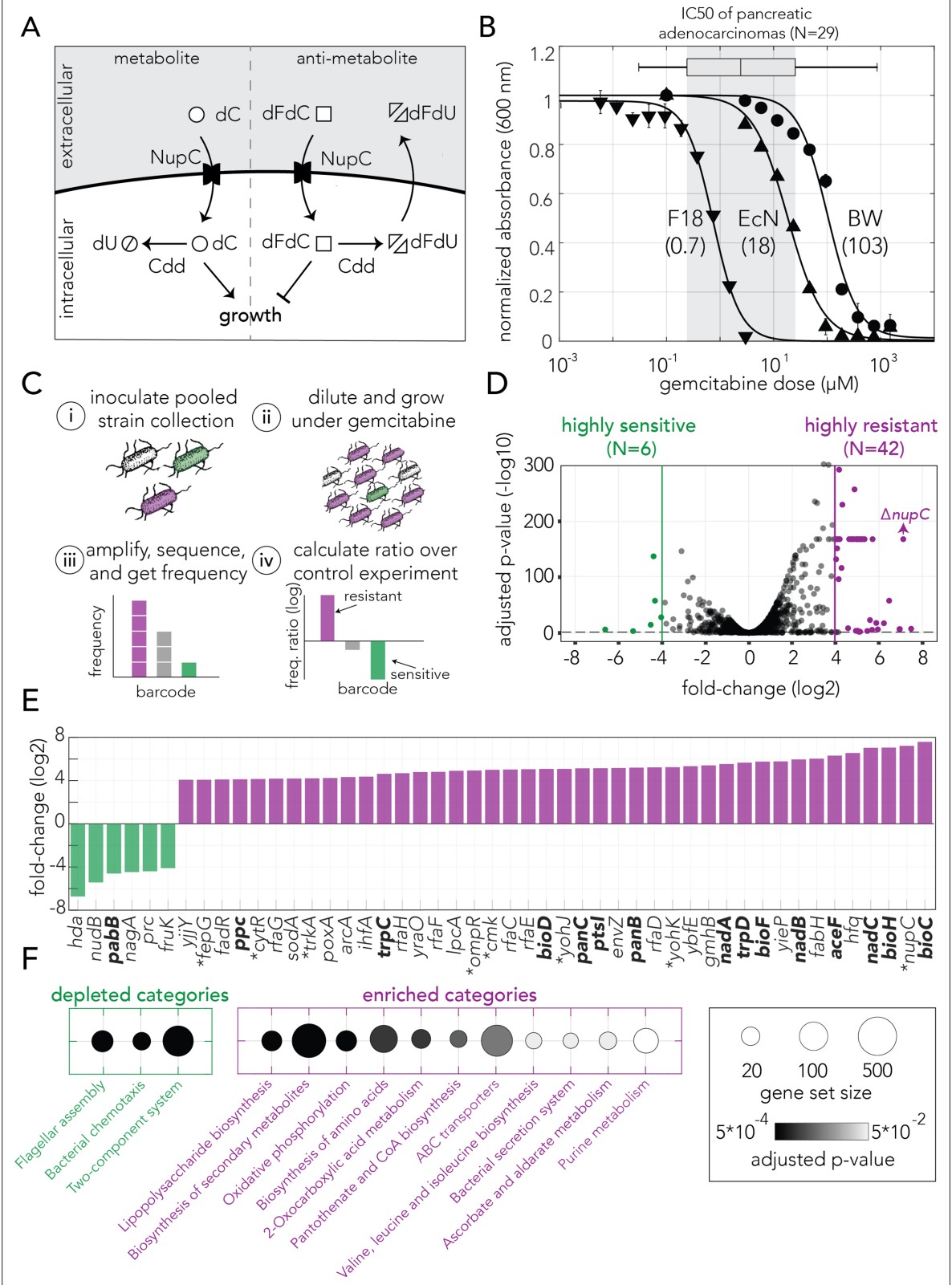

**Figure 1.** Genetic screen identifies gemcitabine sensitive and resistant loss-of-function mutations in *E. coli*. (**A**) Gemcitabine transport and metabolism in *E. coli*. Similar to deoxycytidine (dC), gemcitabine (dFdC) is imported into the cell through the nucleoside permease NupC. Intracellular gemcitabine is either converted into the less toxic metabolite dFdU by the cytidine deaminase Cdd or is phosphorylated and incorporated into the DNA. (**B**) Gemcitabine dose–response curves for three *E. coli* strains (three technical replicates, error bars represent standard deviation, experiment

*Figure 1 continued on next page*

*Figure 1 continued*

performed once). The inferred IC50 are shown in parenthesis. Gemcitabine IC50 range of 29 pancreatic adenocarcinomas are shown as a box plot above the graph. (**C**) Overview of the pooled screening approach. Pooled cultures of the knockout strain collection were inoculated (i) and grown for multiple hours with or without gemcitabine (ii), DNA extracted from cells was used to amplify, sequence and calculate the frequency of barcodes that correspond to individual strains (iii), and the ratio of barcode frequencies of each strain in gemcitabine and control conditions were used to identify sensitive and resistant knockouts (iv). Experiment includes three biological replicates (independent inoculums). (**D**) Volcano plot of the genetic screen results. Green and purple dots represent sensitive and resistant knockout strains, respectively. (**E**) Highly sensitive and resistant strains. Asterisk marks the gene knockouts likely involved in gemcitabine transport and phosphorylation. Knockouts marked in bold were characterized as slow-growing strains. (**F**) Statistically significant enriched and depleted KEGG categories identified by the genetic screen (FDR adjusted p-value cutoff: 0.1). The marker size shows the number of genes in the category and the gray scale marks the statistical significance.

The online version of this article includes the following figure supplement(s) for figure 1:

**Figure supplement 1.** Statistics of the gemcitabine genetic screen performed with the *E. coli* barcoded knockout strain collection.

**Figure supplement 2.** Growth of top five resistant gemcitabine knockouts indentified by the genetic screen in M9 minimal medium.

system for testing how bacterial adaptation can impact drug metabolism and potentially influence the tumor's chemoresistance.

Through a pooled genetic screen, we systematically mapped all loss-of-function mutations that increase *E. coli*'s resistance to gemcitabine and found that inactivation of over 40 genes increased bacterial resistance by more than 16-fold. This observation led us to conclude that resistance can rapidly emerge under natural selection through gene inactivation within a single evolutionary step. Using a functional assay, we found that one third of top resistance mutations impacted extracellular drug concentrations (gemcitabine activity). Co-culturing bacteria harboring these loss-of-function mutations with cancer cells confirmed that these adaptive mutations in bacteria have the potential to alter chemoresistance of neighboring tumor cells. Finally, through in vitro evolution we studied which adaptations emerge under drug selection in three *E. coli* strains. We found that inactivation of the drug transporter NupC arises in all evolved strains. This inactivation leads to decreased bacterial drug import and therefore reduces the rate of gemcitabine breakdown. Reduced bacterial breakdown, in turn, increases gemcitabine availability for neighboring tumor cells. Our work reveals that bacterial adaptation to the frontline chemotherapy drug gemcitabine can take place rapidly in vitro. If similar adaptation takes place in gemcitabine-treated patients, it may ultimately increase the chemosensitivity of the hosting tumor. Our in vitro work suggests that monitoring bacterial adaptation to chemotherapy may be required to decide whether chemotherapy should be augmented with antibiotic treatments. Such decisions are nontrivial given that administration of antibiotics can be detrimental to cancer patients (*Meriggi and Zaniboni, 2021*; *Gao et al., 2020*; *Corty et al., 2020*; *Elkrief et al., 2019*).

## Results

### The *E. coli* resistome against gemcitabine

We first set out to determine the inhibitory concentration of gemcitabine in three *E. coli* strains: BW25113 (a K-12 lab strain), F-18 (a human fecal isolate), and Nissle 1917 (a human fecal isolate that is used as probiotic). We characterized the inhibitory concentrations by monitoring bacterial growth inhibition after 12 hr. *Figure 1B* shows the sensitivity curves and the inhibitory concentration that reduced culture density by 50% (IC50). We observed that gemcitabine can completely inhibit the growth of all three strains, with F-18 being most sensitive (IC50 = 0.7 μM) and BW25113 being most resistant (IC50 = 103 μM). These concentrations are comparable to the IC50 reported for 29 pancreatic adenocarcinomas in the GDSC2 dataset that tested hundreds of toxic compounds on a thousand cell lines (*Yang et al., 2012*; boxplot above *Figure 1B*).

We next conducted a genetic screen with a collection of 3680 single-gene knockout strains to systematically map all nonessential genes that influence gemcitabine resistance when depleted (typically referred to as the drug resistome). We used a pooled screening approach that we recently developed, which relies on sequencing DNA barcodes that are unique for each gene-knockout in the strain collection (*Rosener et al., 2020*; *Noto Guillen et al., 2021*). *Figure 1C* outlines the main steps of the screening method: the pooled knockout strains were inoculated and grown in media containing a high gemcitabine concentration (140 μM) while a control culture was inoculated in media without drug.

Once the cultures reached a late logarithmic growth phase, cells were lysed, and DNA was extracted for amplification and sequencing of the barcode region. Lastly, we compared the frequency of each barcode, corresponding to an individual gene-knockout, in the drug and control conditions. This comparison revealed strains whose frequency was significantly increased or decreased in gemcitabine. Such enrichment or depletion correspond to increased or decreased drug resistance, respectively. We used three biological replicates, biological experiments conducted side-by-side, in three separate tubes, to infer statistical significance. We identified over a million and a half barcode containing reads in each replicate that corresponded to roughly 3500 unique barcodes (knockout strains). *Figure 1—figure supplement 1* shows detailed information on the screen coverage and replication quality. To validate the screen results we repeated the entire screen two additional times, once with the same collection of knockout strains and once with a collection knockout strains that were cloned independently (again, each with three internal replicates). The results (resistance/sensitivity) from the two additional screens were highly correlated with the original screen (*Figure 1—figure supplement 1D*). Results of all screens and replicates appear in *Supplementary file 1*.

*Figure 1D* shows a volcano plot representation of the screen results. Using very strict cutoff values for fold-change (>16) and false discovery rate (FDR)-adjusted p-value (<0.05), we identified 42 gemcitabine-resistant knockout strains and 6 gemcitabine-sensitive strains (shown as purple and green circles in *Figure 1D*). The screen results, including all statistically significant hits without a strict cutoff on enrichment, appear in *Supplementary file 1*. Reassuringly, we found that knockout of the known drug transporter (*nupC*) is among the top resistors. When we inspected the gene annotation of the top resistors, we identified multiple hits from the known target pathway of the drug (*Figure 1E*). These included the permease (*nupC*), the transcriptional regulator *cytR* (a repressor of both *nupC* and *cdd*), and the cytidylate kinase (*cmk*) that likely phosphorylate intracellular gemcitabine. We also found multiple hits that encode membrane proteins or transporters, including *ompR* and *envZ* that together regulate permeability channels for nutrients, toxins, and antibiotics (*Mizuno and Mizushima, 1987*). Lastly, we observed a high prevalence of genes coding for metabolic enzymes that can considerably slow down growth when mutated (*Noto Guillen et al., 2021*). Indeed, a statistical test revealed that gene knockouts that were previously identified as reducing growth (*Baba et al., 2006*) were highly enriched in the set of gemcitabine resistant strains (p-value = 8.6186e-18, Fisher's exact test). The overlap between resistant strains and slow growing strains is sensible given that slow growth reduces the rate of DNA synthesis. Since gemcitabine is quickly degraded by the bacterial Cdd enzyme, it was only transiently present in the extracellular media during the screen experiment. Under such a transitory stress, slower consumption of the antimetabolite is likely beneficial (normal growers incorporated much more gemcitabine into their DNA and remain arrested while slow growers incorporated less and therefore avoid arrest).

Next, we used the gene set enrichment analysis tool GAGE (*Luo et al., 2009*) to test for functional enrichment using the Kyoto Encyclopedia of Genes and Genomes (KEGG) (*Kanehisa and Goto, 2000*) and Gene Ontology (GO) (*Ashburner et al., 2000*) databases. This analysis is complementary to the previous analysis since it considers the enrichment values from all strains rather than only the limited set of hit strains obtained by imposing strict cutoffs. *Figure 1F* shows the functional enrichment by KEGG pathways (enrichment indicates that pathway inactivation increased resistance). Reassuringly, we observed a high agreement between the functional enrichment by GAGE and the annotation of the top hits. Specifically, we observed enrichment in purine synthesis and membrane transporters as well as multiple metabolic pathways impacting bacterial growth rate (*Noto Guillen et al., 2021*). *Supplementary file 2* provides the full list of enriched and depleted categories.

Taken together, the results from our genetic screen outline three adaptation strategies that increase bacteria's gemcitabine resistance: reduced drug import by inactivating membrane proteins and transporter systems, changes in the drug metabolism through mutations in the target pathway, and inactivation of metabolic genes that slowdown bacterial growth. Importantly, since these resistance adaptations arise from knockout of single genes, they are all accessible within a single evolutionary step (e.g., a single gene inactivating mutation).

## The impact of bacterial resistance on bacterial drug degradation

Our genetic screen revealed alternative adaptation strategies that increase bacterial resistance against gemcitabine. However, for most loss-of-function mutations, it remains to be determined how

they will influence the rate of bacterial drug degradation (and ultimately drug availability for neighboring cancer cells). We therefore designed a functional assay to detect changes in drug activity, after bacterial incubation with the drug, relative to its activity after incubation with the wild-type strain (outlined in *Figure 2A*). We incubated a knockout strain of interest in saline with a high gemcitabine concentration and collected the conditioned supernatant after a short incubation period (15 or 45 min). We diluted the conditioned supernatant into regular media and monitored the growth of a drug-sensitive reporter strain (a *cdd* knockout that cannot degrade gemcitabine) in this media. Finally, we used growth curves of the reporter strain as a proxy for the gemcitabine concentration in the conditioned supernatant. We reasoned that a conditioned supernatant containing high drug concentration is indicative of slow degradation by the strain of interest. This difference in drug concentration will in turn manifest as slow growth of the reporter strain (blue curve in *Figure 2A*). We note that this detection method is insufficient to resolve the mechanism underlying the slowed degradation since it only measures drug availability in the extracellular environment after incubation (e.g., both slow import and slow deamination will be interpreted as slow degradation).

We evaluated drug activity after incubation of the 88 most resistant strains, the top 42 resistors found by the strict enrichment cutoff (*Figure 1E*) and the next 46 resistant strains (supernatant collection was repeated three times on different days). *Figure 2B* shows a summary of the results of this assay (*Figure 2—figure supplement 1* shows observed growth curves). We found one third of the tested knockout strains modulated extracellular drug availability (33 of 88). Specifically, 11% of all strains were fast degraders and 26% of strains were slow degraders (one-tailed *t*-test, FDR-adjusted p-value<0.1). We next decided to validate the conclusions of our functional assay using an independent chemical approach for the wild-type and three knockout strains. We focused on the fastest and the slowest degraders and cdd knockout as a control. We incubated each strain with gemcitabine and sampled aliquots of the supernatant at predetermined timepoints (20, 45, and 75 min). We then used gas chromatography–mass spectrometry (GC-MS) to measure the concentration of gemcitabine and its degradation product in the conditioned supernatant (*Figure 2C*). In agreement with our functional assay, the GC-MS measurements confirmed that the gene knockouts indeed altered the availability of gemcitabine and in the extracellular environment. The increased availability of the drug breakdown product (dFdU) support the conclusion that rate of drug metabolism is underlying this change (as opposed to intracellular drug accumulation; *Klünemann et al., 2021*).

Lastly, we tested whether bacterial resistance mutations influence the drug sensitivity of co-cultured cancer cells. While previous works relied on sequential exposure to the drug (*Geller et al., 2017*; *Lehouritis et al., 2015*), using conditioned bacterial supernatant on cancer cells, we reasoned that a co-culture system will reveal whether bacterial degradation is sufficient to impact drug efficacy in neighboring cancer cells that are simultaneously exposed to the drug. We co-cultured bacteria with spheroids of the CT26 murine cancer cell line and simultaneously treated them with gemcitabine for 4 hr (gemcitabine cytotoxicity can be recapitulated in this cell line; *Geller et al., 2017*). Spheroids were then washed to remove bacteria and left to grow for a week in media supplemented with antibiotics. Lastly, spheroids were washed again to remove dead cells and the area of the spheroids was measured to evaluate gemcitabine's efficacy on the cancer cells. *Figure 2D* shows microscopy images of a representative microwell plate. As the figure shows, we observed sensible trends in these experiments: first, we observed that spheroid size was inversely correlated with gemcitabine concentration (lower spheroid row in *Figure 2D*) and that bacterial concentration, without any drug, did not significantly impact spheroid growth (left spheroid column in *Figure 2D*). Reassuringly, we observed that co-cultured bacteria can mitigate gemcitabine damage and that the magnitude of rescue depended on the bacterial concentration (spheroid area is overall increased in upper spheroid rows in *Figure 2D*).

We used our systematic measurements of spheroid area to fit a fitness landscape. This procedure allowed us to infer the effective drug concentration that leads to a 50% change in area (EC50) for any bacterial concentration. *Figure 2E* shows the landscapes for the wild-type strain, a control non-degrader (*cdd* knockout) and the fastest and slowest degraders (*cytR* and *nupC knockouts*, respectively). As evidenced by the changes in the EC50 fronts, we observed that the spheroid fitness landscapes were considerably different depending on the co-cultured bacteria during drug exposure. In agreement with our functional and chemical assays, increased chemoresistance relative to the wild-type strain was observed when spheroids were co-incubated with the fast degrader (*cytR* knockout). In contrast, decreased chemoresistance relative to the wild-type strain was for co-incubation was with

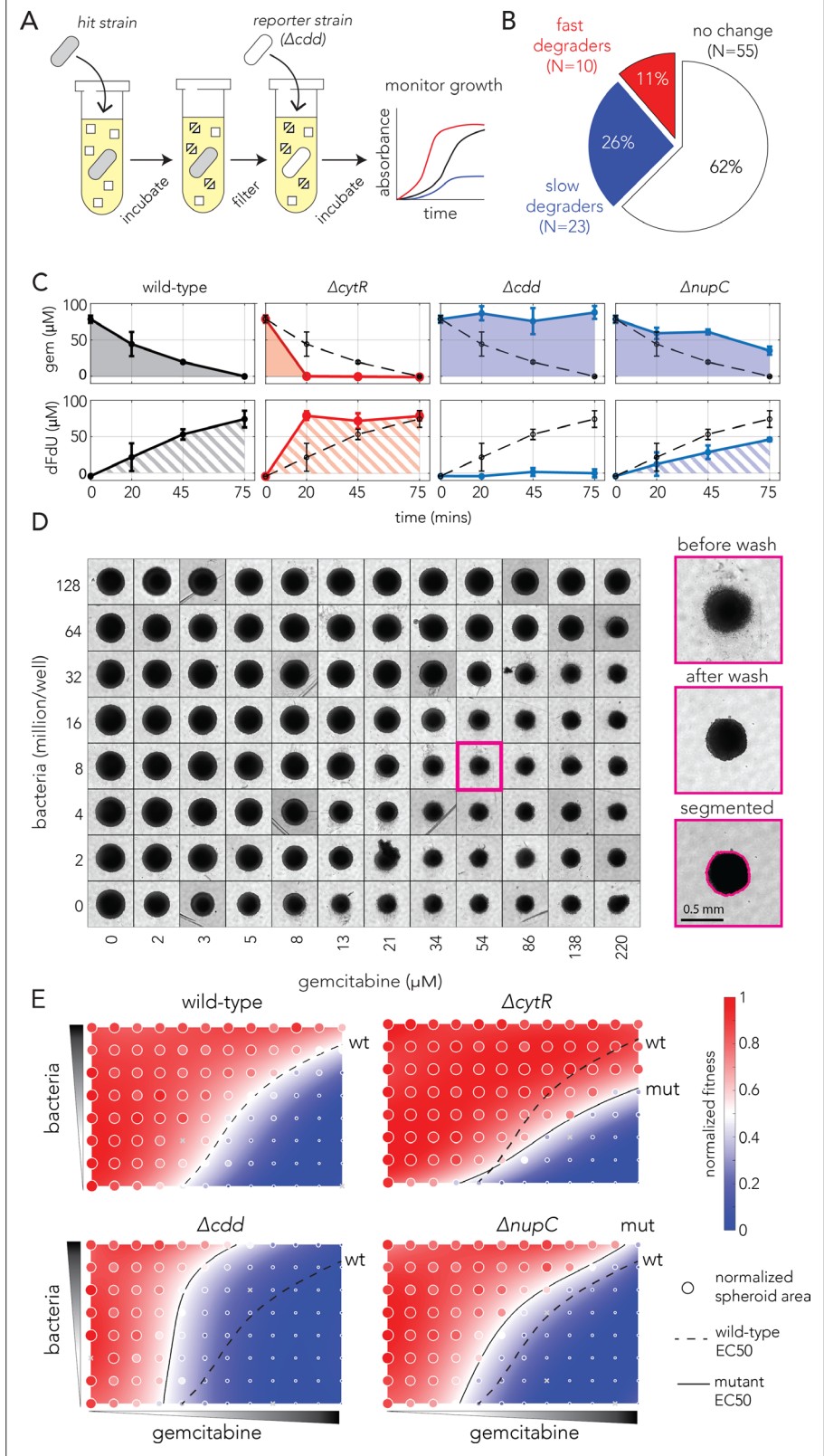

**Figure 2.** Bacterial gemcitabine resistance can oppositely affect drug degradation and impact neighboring cancer cells. (**A**) The functional drug breakdown assay. Each knockout strain was inoculated in saline containing gemcitabine and incubated for 15 or 45 min. Conditioned supernatant was then filtered and mixed with fresh media before inoculating a reporter strain. Reporter strain growth was used as a proxy for changes in gemcitabine

*Figure 2 continued on next page*

*Figure 2 continued*

availability in the conditioned supernatant. (**B**) Results of the drug breakdown assay for the top 88 resistant knockouts. A third of all tested knockouts (33/88) influenced the drug degradation rate. (**C**) Chemical validation of the functional assay for the slowest and fastest degraders. GC-MS was used to measure gemcitabine concentration (shaded area, top panels) and its degradation product dFdU (hatched area, lower panels) in conditioned supernatant (colors as in **A**). The dashed black curve marks the measurements after incubation with the wild-type strain. The error bars show the standard deviation of three biological replicates. (**D**) Co-culture experiments shows bacterial mutations can influence gemcitabine efficacy in neighboring spheroids of cancer cells. Representative microscopy images of spheroids that were co-cultured with wild-type bacteria across multiple concentrations of drug and bacteria (large images show the same spheroid before the wash, after the wash that removed dead cell debris, and after image segmentation). (**E**) Bacterial mutations that modulate drug degradation can impact drug efficacy in neighboring cancer cells. Results of spheroid experiments with the slowest and fastest degraders and the inferred fitness landscapes. Each panel shows the fitness landscape calculated from spheroid size across a range of gemcitabine and bacteria concentrations (as in **D**). The color code shows the normalized spheroid size (ranging from the smallest spheroid to the largest one). The dashed black line marks the parameter combination (bacteria and gemcitabine concentrations) that reduce spheroid growth by 50% (EC50 front), and solid lines show the EC50 front for the knockout strains. Shifts in the EC50 front, relative to the wild-type front, indicate that co-incubation with the knockout strain during drug exposure altered the spheroid's chemoresistance.

The online version of this article includes the following figure supplement(s) for figure 2:

**Figure supplement 1.** Individual growth curves of reporter strain (*cdd* knockout) in the functional assay to estimate gemcitabine breakdown rate of the top 88 gemcitabine resistant genetic screen hits.

**Figure supplement 2.** Validation of results of spheroid experiment (***Figure 2E***).

**Figure supplement 3.** Changes in area under the EC50 front in spheroid fitness landscapes generated with co-incubation of gemcitabine and selected gemcitabine-resistant knockout strains (***Figure 2E***).

---

the slow degrader (*nupC* knockout). A replicated spheroid experiment with *cytR* and *nupC* knockouts and the wild-type strain revealed similar shifts in the EC50 fronts (***Figure 2—figure supplement 2***). Additional experiments with nine more bacterial resistors showed they also impacted spheroid chemoresistance (***Figure 2—figure supplement 3***).

Taken together, the results of the spheroid experiments demonstrate that mutations impacting gemcitabine degradation rates in bacteria can indeed impact neighboring cancer cells simultaneously with the drug. Importantly, we observed that bacterial resistors can have opposite influences on gemcitabine sensitivity of co-cultured cancer cells. For examples, the *nupC* knockout decreased chemoresistance while the *cytR* knockout increased it. A key question remaining is which adaptations will naturally transpire during bacterial evolution under drug selection.

## Evolved bacterial resistance against gemcitabine

The genetic screen uncovered multiple loss-of-function mutations that confer bacterial gemcitabine resistance. Yet, such screens are insufficient for determining which gene inactivation, if any, will emerge under natural selection. Moreover, since evolution can leverage additional processes beyond gene inactivation, such as gain-of-function, adaptation may follow an entirely different evolutionary trajectory. We applied the widely used serial transfer approach to select for evolved drug resistance in bacteria (***Dragosits and Mattanovich, 2013***). Such in vitro experiments can shed light on the mechanisms underlying resistance and the time scale needed to acquire resistance. To test whether reoccurring adaptations emerge, we used the three *E. coli* strains that were characterized by different drug sensitivity levels (***Figure 1B***). Once the serial transfer experiment ended, we evaluated whether drug resistance increased in the population and isolated single-resistant clones for whole-genome sequencing (***Figure 3A***).

We evolved three independent populations of each *E. coli* strain in sub-inhibitory concentrations of gemcitabine ('Materials and methods') and three control populations without any drug. We monitored drug resistance in all populations daily throughout the experiment (***Figure 3B***). We observed that resistance emerged within a day or two for the Nissle 1917 and F-18 strains, while it emerged more slowly for the BW25113 strain. Last day populations from all strains that evolved in gemcitabine were resistant to the drug across all tested drug concentrations (***Figure 3C***). In order to isolate individual resistant clones, we streaked each of the evolved population on agar plates and measured IC50 dose for eight independent colonies. (***Figure 3D***). Almost all clones from Nissle 1917 and F-18

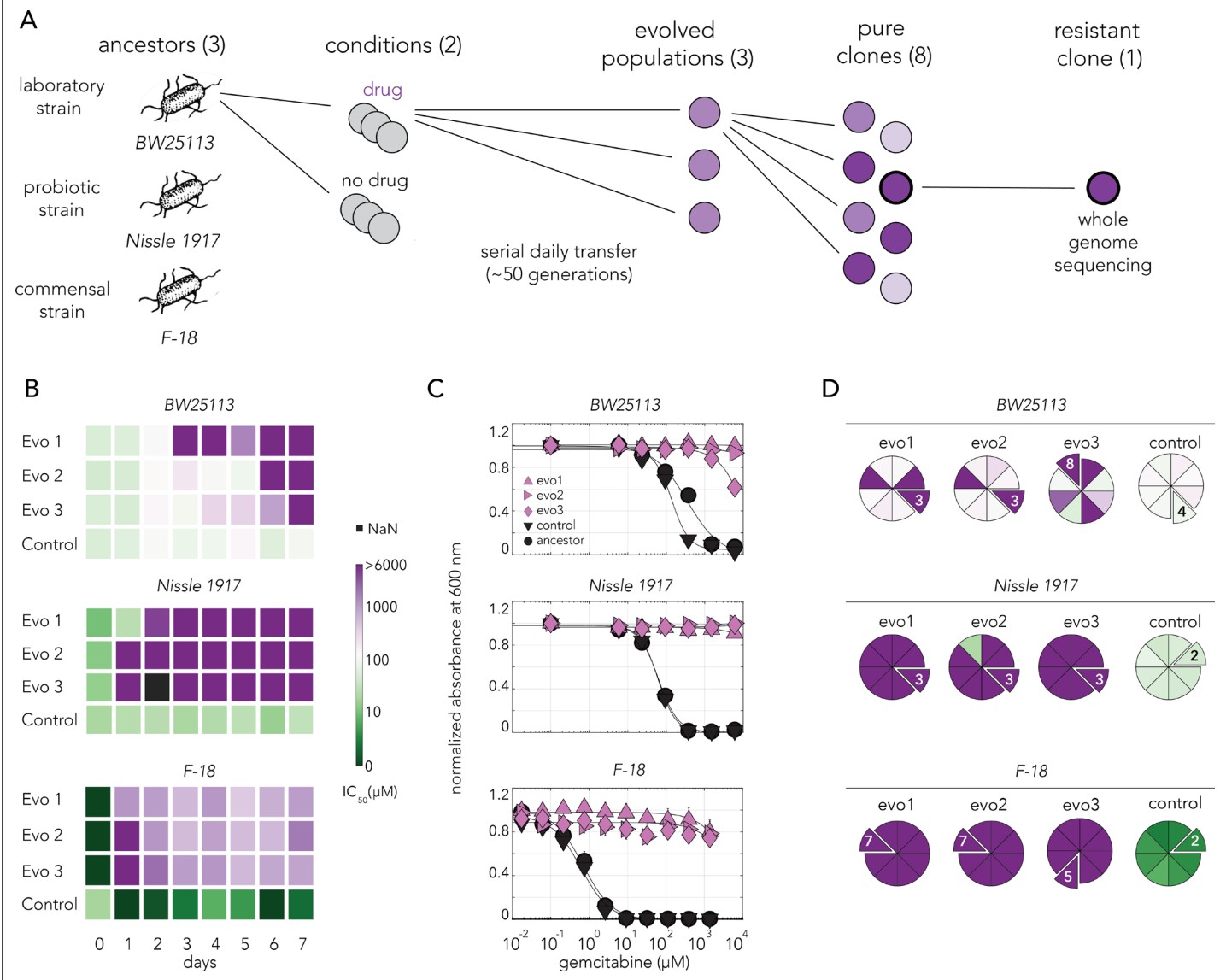

**Figure 3.** Gemcitabine selection leads to rapid evolved resistance in three *E. coli* strains. (**A**) Overall approach for the lab evolution experiment. Three *E. coli* strains were evolved over 50 generations in serial transfer evolution experiment. We measured gemcitabine resistance of last day populations and eight single clones were isolated from each population. A single resistant clone was used for the whole-genome sequencing and for identifying of the underlying mechanism for drug resistance. (**B**) Heatmaps showing the temporal changes in gemcitabine IC50 of evolving populations in each day of the serial transfer experiment. (**C**) Gemcitabine dose–response curves of last day populations from lab evolution experiment. All of the gemcitabine evolved lines, but not the control evolved lines, developed a high resistance against gemcitabine. Error bars show the standard deviation of three technical replicates. (**D**) Pie charts showing the gemcitabine IC50 levels of screened clones. Slices represent the clone selected for whole-genome sequencing. Color scale is the same as panel (**B**).

drug evolved strains were resistant. However, clones from the BW25113 populations showed heterologous levels of resistance. All clones isolated from the populations that evolved without the drug were gemcitabine sensitive. We chose a single clone from each population for further analysis (marked as extruding slices in the pie charts in *Figure 3D*). These individual clones represent lineages that evolved completely independently from one another.

## Inactivation of *nupC* underlies evolved drug resistance

Phenotypic measurements revealed that all drug-evolved populations became highly resistant. We next sequenced the genomes of evolved clones to identify the underlying adaptive mutations and identified mutations with the BreSeq software (*Barrick et al., 2014*). *Figure 4A* shows the mutations

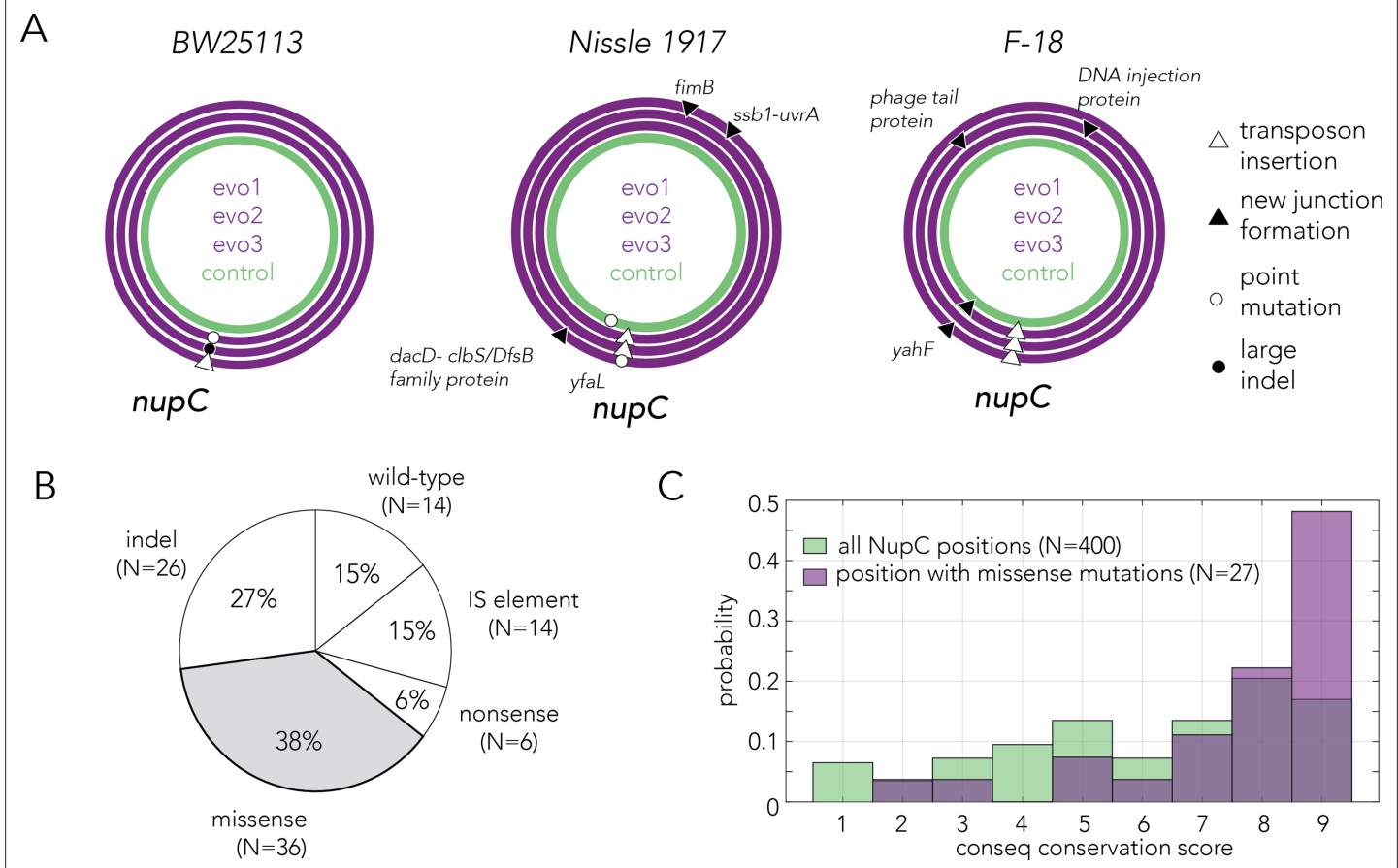

**Figure 4.** Evolved resistance converges to inactivation of the nucleoside permease NupC. (**A**) Circa plots showing the mutations identified by whole-genome sequencing in pure clones isolated from independently evolved populations. Mutations in the coding region of the *nupC* gene were observed across all gemcitabine-evolved clones but not in the no-drug control evolved clones. Since the F-18 genome is not fully assembled, relative positions on the F-18 circa plot are not real genomic positions. (**B**) Pie chart showing the frequency of various mutation types identified by sequencing the *nupC* gene across 96 spontaneous gemcitabine resistant mutants in the BW25113 strain. (**C**) A comparison of the evolutionary conservation of missense mutation positions relative to the conservation of all positions in the *nupC* gene. The positions of the missense mutations are statistically biased towards the conserved positions (p-value = $1.95 * 10^{-4}$ in Wilcoxon rank-sum test).

The online version of this article includes the following figure supplement(s) for figure 4:

**Figure supplement 1.** Annotation of *nupC* mutations by evolutionary conservation.

**Figure supplement 2.** Plots showing the calculated mutation rates for the loci *nupC* and *nfsA* in four *E. coli* strains using Luria–Delbrück fluctuation experiments.

we identified in the single clones as circa plots (concentric circles representing the bacterial chromosomes). Annotation of mutations in F-18 required careful manual inspection since its reference genome consists of 113 contigs. As the figure shows, we observed that the genomes of all the drug-evolved clones harbored *nupC* mutations while none of the control-evolved clones had such mutations. **Supplementary file 3** provides the BreSeq information of mutations we identified. Importantly, while we found additional mutations beyond those in the *nupC* gene, we did not observe another gene that was repeatedly mutated across all independent replicates in all of the *E. coli* strains. The only other repeated mutation we identified involved the *yahF* gene in two lines of F-18 that also shared an identical *nupC* mutation (transposon). Inspecting the BreSeq report and the contig files led us to believe both lines shared single new junction that impacted both the *nupC* and *yahF* loci.

The mechanism of mutation in the *nupC* gene varied across the gemcitabine-evolved strains and included point mutations, large deletions, and new junction formation within *nupC* coding region that can stem from transposon insertions and genomic rearrangements. Two of the point mutations we identified were missense mutations (S175P and V249A). To evaluate whether these point mutations

likely interfere with the permease function, we used the ConSurf Server that identifies evolutionarily conserved positions (*Ashkenazy et al., 2016*). The analysis revealed that both positions are highly conserved and are therefore likely important for the permease function (positions marked in red in *Figure 4—figure supplement 1*). Lastly, we examined the function of all other genes that were mutated in evolved strains to pinpoint additional putative adaptive mutations. We identified a new genomic junction, likely originating from a transposon insertion, upstream to the *uvrA* gene. The *uvrA* gene codes for A subunit in the UvrABC nuclease that is involved in the nucleotide excision repair pathway (*Keseler et al., 2017*). The mutation impacted the annotated promoter of the gene according to regulonDB (*Gama-Castro et al., 2016*). Taken together, we concluded that resistance emerged across all 12 sequenced and independently gemcitabine-evolved strains primarily through inactivation of the nucleoside permease NupC.

## Mechanisms underlying convergence towards *nupC* inactivation

Gemcitabine adaptation in our evolution experiments likely emerged through inactivation of *nupC* across all evolved strains. This convergence can be driven by multiple mechanisms that are not necessarily exclusive to one another. Evolutionary trajectories are influenced by multiple parameters, including the adaptation benefit (e.g., the level of resistance the mutation confers), the adaptation cost (e.g., if it reduces growth), and the likelihood that the mutation will appear spontaneously. We reasoned that quantifying these parameters for the *nupC* gene would provide insight into the forces underlying the evolutionary convergence we observed.

Our genetic screen already revealed that *nupC* is among the top loss-of-function mutations conferring resistance (*Figure 1E*), yet it remains unclear whether *nupC* inactivation is associated with any cost to the cells. We therefore monitored the growth rate of the top five resistant knockouts identified in the genetic screens (*Figure 1—figure supplement 2*). The experiment revealed that a *nupC* knockout grows as fast as the wild-type strain (*ybcN* knockout) for most of the growth phases and slows down only during the stationary phase. In contrast, all other top resistant knockouts were significantly slower than the wild-type strain (and *nupC*) during all growth phases. We concluded that *nupC* inactivation is unique among the resistant mutations since its associated with only a small fitness cost for bacteria when the drug is not present.

We next tested whether the *nupC* locus contains a mutation hotspot. We reasoned that a mutation hotspot will be evident if we observed that multiple independent resistant clones will share a specific site or region within the *nupC* locus or will leverage on a specific mutation mechanism. We obtained independent clones by picking 100 individual colonies of the wild-type strain. We grew them overnight and plated them on agar plates with an inhibitory gemcitabine concentration (0.5 mM). We reasoned that at this high concentration, only the most resistant and fast-growing mutants, namely *nupC* mutants, will be able to form large colonies within an overnight growth. We then picked a single resistant colony from each of the 100 agar plates and Sanger sequenced the *nupC* coding sequence locus. We found that 85% of colonies were mutated in this region, with mutations spanning multiple types. Missense mutations and short indels were the most frequent mutations observed (*Figure 4B*). In order to annotate the missense mutations, we tested whether they likely disrupt positions important for permease function. We used the ConSurf sequence analysis tool and identified evolutionarily conserved positions (*Figure 4—figure supplement 1*). The analysis revealed that missense mutations we identified were significantly biased towards the highly conserved regions of the permease (*Figure 4C*, p-value = $1.95 * 10^{-4}$ in Wilcoxon rank-sum test).

Lastly, we decided to test whether the *nupC* is naturally poised for frequent mutations in the absence of gemcitabine. We reasoned that since our agar plating experiment showed that most resistant colonies have been *nupC* mutants, we could estimate the mutation rate (μ) of in this gene with a Luria–Delbrück fluctuation test (*Luria and Delbrück, 1943*). We compared this mutation rate to a reference gene (*nfsA*) that confers resistance to furazolidone (*Lourenço et al., 2016*). We performed this experiment as four replicates using four different genetic backgrounds (the strains used for the evolution experiments and the MG1655 lab strain). *Figure 4—figure supplement 2* and *Supplementary file 4* shows the calculated mutation rates for *nupC* and *nfsA* loci in all strains (normalized to a gene with 1000 basepairs). In all cases, we found rates comparable to the average gene mutation rate previously inferred for *E. coli* ($2.1 \times 10^{-7}$ per gene per generation) (*Chen and Zhang, 2013*). For BW25113, the mutation rate of *nupC* was 5.7-fold higher than that of *nfsA*. For Nissle, this difference

was 6.8-fold and for MG1655 it was 1.8-fold. For F-18 strain, in contrast, the mutation rate of *nfsA* was 7.9-fold higher than *nupC*. While the mutation rate was not identical in the two genes, the mutation of *nupC* was not always higher than that of *nfsA*, and their rate difference was never larger by an order of magnitude.

Taken together, the results from multiple experiments suggested that *nupC* was likely repeatedly inactivated across multiple genetic backgrounds since it confers high drug resistance without compromising growth. Experiments focusing on the *nupC* mutation revealed that gene inactivation could take place through multiple alternative mutation mechanisms and that the *nupC* locus is not characterized by an exceptionally high mutation rate (excluding the possibility of a mutational hotspot).

## Discussion

The recent discovery that bacterial infections are frequent across multiple cancer types suggests that the tumor-microbiome is an important, yet understudied, component of the tumor microenvironment (*Cullin et al., 2021*; *Nejman et al., 2020*). Here, we suggest that a key aspect of microbial biology is underexplored in current investigations of the tumor-microbiome – the ability of bacteria to rapidly evolve and adapt to extracellular changes. Within the tumor niche, successful colonization may require bacterial adaptation to the unique conditions of the tumor microenvironment, including adaptation to tumor-targeting therapeutics. A strong selective pressure from chemotherapies likely exists given that multiple antineoplastic drugs are putative antimicrobials at physiological concentrations (*Maier et al., 2018*). We previously raised the hypothesis that bacterial evolved resistance to tumor-targeting chemotherapies can manifest in changes to bacterial drug metabolism and this type of adaptation can inadvertently influence chemoresistance. Our previous work used the *Caenorhabditis elegans* model system, its bacterial diet, and two fluoropyrimidine chemotherapy drugs to study this hypothesis (*Rosener et al., 2020*). In that model system, we estimated that almost 60% of loss-of-function mutations that conferred bacterial resistance would also reduce drug toxicity in a worm host feeding on these bacteria. Here, we further study this hypothesis and focus on a different chemotherapy drug and a model system that captures bacterial–drug interactions that may be at play in the tumor microenvironment, rather than in the host gut as in the previous publication.

Research of the bacterial role in pancreatic cancer revealed multiple and independent mechanisms of microbial influence on cancers in this organ (*Nejman et al., 2020*; *Riquelme et al., 2019*; *McAllister et al., 2019*; *Geller et al., 2017*; *Aykut et al., 2019*; *Pushalkar et al., 2018*). Specifically, recent work suggested that pancreatic colonization by proteobacteria can decrease efficacy of gemcitabine through rapid bacterial drug inactivavation (*Geller et al., 2017*). We used this drug–bacteria–tumor interaction to test our hypothesis by reconstituting similar interactions from individual parts that are well-understood on their own. We first mapped the gemcitabine resistome in *E. coli* and found that changes in multiple cellular processes increase resistance. Importantly, these potential adaptations are easily accessible within short time scales since they only require inactivation of a single gene. A functional assay revealed that one third of these adaptations impact bacterial drug breakdown (*Figure 2B*). Indeed, gemcitabine administration to co-cultures of cancer spheroids and bacteria demonstrated that two of the top loss-of-function mutations that the screen identified can considerably, and oppositely, impact drug efficacy in neighboring cancer cells (*Figure 2E*). While the impact of bacterial resistance on neighboring cells could have been predicted for the *nupC* knockout, it was not trivial for the *cytR* knockout. CytR is a transcription factor that represses at least 14 different operons, including nucleotide transporters and membrane proteins (*nupC*, *nupG*, *tsx*, *ycdZ*), sigma factors (*rpoH*), and metabolic enzymes in the target pathways (*ccd*, *udp*, *deoABCD*) (*Keseler et al., 2017*). A chemical assay revealed that *cytR* knockdown culminates in increased drug degradation and a co-culture experiment confirmed it increases chemoresistance of neighboring cancer spheroids. It is therefore likely that relief of CytR repression of both *nupC* and *cdd* leads to increased import that is counteracted by even faster gemcitabine deamination. This observation is key since it suggests that mutations conferring gemcitabine resistance in bacteria can both increase and decrease gemcitabine breakdown rate. In the context of the tumor-microbiome, bacterial adaptation to gemcitabine can therefore raise or reduce the tumor's chemoresistance.

A key question that arises from the observation that bacterial resistance can modulate drug availability for neighboring cancer cells, is which adaptations will emerge under drug selection and whether evolution will repeatedly converge to the same resistance mechanism. Results from the serial transfer

evolution experiment in three *E. coli* strains showed that drug selection repeatedly yielded adapted clones that disabled the drug permease (*Figure 4A*). Additional experiments suggested that selection for *nupC* inactivation, at the expense of alternative resistance mechanisms, is attributed to the high resistance this inactivation confers and to the minimal impact it has on growth. Importantly, as the fluctuation experiments reveal, resistant clones can preexist drug exposure and therefore also likely preexist any treatment if the bacterial number in the tumor-microbiome is sufficiently large. In such cases, tumor-microbiome adaptation can potentially take place through ecological-like changes and takeover by a resistant bacterial clone. Such clonal expansion in bacteria is reminiscent to the process of clonal expansion and takeover of preexisting resistant cancer cells that is thought to be prevalent in cancer treatment.

Previous research established that the microbiomes, in natural body sites or within tumors, can metabolize tumor-targeting drugs and by doing so influence drug efficacy in the host. Our previous and current studies complement this premise by demonstrating that bacterial evolutionary adaptation to chemotherapies can further influence bacterial drug metabolism and therefore host drug efficacy. Intriguingly, in this work we found that bacterial influence can have opposite effects on drug break-down and therefore can increase, or decrease, chemoresistance. Moreover, the genetic screens from both works suggest that such occurrences might not be rare, given that such a high fraction of adaptive loss-of-function mutations also alter drug breakdown rates.

While we believe this work illuminates an underexplored and potentially impactful field of research – evolutionary adaption in the tumor microbiome – we are also excited by the follow-up questions that naturally ensue our in vitro findings. Foremost, it will be fascinating to test whether tumor-colonizing bacteria evolve and adapt to anticancer host treatment in animal models. Specifically, following our observations in the model gammaproteobacteria *E. coli*, and the known prevalence of gammaproteo-bacteria colonization in pancreatic cancer (*Geller et al., 2017*), it will be intriguing to check whether similar adaptations are observed in bacteria isolated from tumors that were resected from pancreatic cancer patients. Identifying convergent adaptation in tumor-microbiome of patients may prove impactful for personalizing anticancer treatment and informing the decision to complement chemo-therapy treatment with antibiotics, a decision that is highly consequential for cancer patients and that should therefore be well-justified (*Meriggi and Zaniboni, 2021*; *Gao et al., 2020*; *Corty et al., 2020*; *Elkrief et al., 2019*).

## Materials and methods
### Bacterial strains and growth conditions
Bacterial strains used in this study are shown in *Table 1*. We used the *E. coli* barcoded knockout strain collection for the pooled genetic screen (similarly to *Rosener et al., 2020*; *Noto Guillen et al., 2021*). All experiments measuring gemcitabine breakdown were performed with strains from the KEIO strain collection (*Baba et al., 2006*). For spheroid experiments, double knockout strains were generated with P1 transduction method (*Thomason et al., 2007*) using *pyrD* knockout strain from the barcoded library and the desired gene knockout from the KEIO collection. The *pyrD* knockout background was used since it is a pyrimidine auxotroph that cannot grow in the media used to culture the spheroids.

For all experiments, bacteria were inoculated into Lysogeny Broth (LB) and grown overnight at 37°C, 200 rpm orbital shaking. Knockout strains were grown in LB media supplemented with 50 µg/mL kanamycin (KEIO strain collection) or 25 µg/mL chloramphenicol (barcoded strain collection). All growth and serial evolution experiments were performed in M9 minimal media supplemented with 0.2% amicase and 0.4% glucose. Experiments designed to monitor gemcitabine breakdown were performed in PBS (functional assay) or in 0.9% saline (GC-MS).

### Barcoded strain library
The *E. coli* barcoded deletion library was used as in our previous studies (*Rosener et al., 2020*; *Noto Guillen et al., 2021*). The parent strain of this library is BW38028 with the genotype Δ(araD-araB)567 lacZp-4105(UV5)-lacY hsdR514, rph+ (*Conway et al., 2014*). The library includes two sets of 3680 knockout strains (each set with different barcodes: odd and even libraries). After overnight growth in nutrient-poor synthetic media (M9), we identified 3512 barcodes in one set and 3226 barcodes in the other set (3145 knockout strains were detected in both collections). In each strain, the open-reading

**Table 1.** Bacteria used in this study.

| Strain | Source | Remarks |
|---|---|---|
| *E. coli* BW25113 | Walhout Lab, University of Massachusetts Chan Medical School, MA, USA | |
| *E. coli* F-18 | McCormick Lab, University of Massachusetts Chan Medical School, MA, USA | |
| *E. coli* MG1655 | Brewster Lab, University of Massachusetts Chan Medical School, MA, USA | |
| *E. coli* Nissle 1917 | ArdeyPharm GmbH, (Pharma Zentrale GmbH), Germany | |
| *E. coli* KEIO knockout collection | Dharmacon (GE Life Sciences) | |
| *E. coli* Barcoded knockout collection (barcoded library) | Hirotada Mori, Nara Institute of Science and Technology, Japan | Parent Strain: BW38028 |
| BW25113 Δ*pyrD*::tGFP-chl$^r$-barcode Δ*cdd*::kan$^r$ | Barcoded library, KEIO Collection | Double knockout was generated by P1 transduction |
| BW25113 Δ*pyrD*::tGFP-chl$^r$-barcode Δ*cytR*::kan$^r$ | Barcoded library, KEIO Collection | Double knockout was generated by P1 transduction |
| BW25113 Δ*pyrD*::tGFP-chl$^r$-barcode Δ*envZ*::kan$^r$ | Barcoded library, KEIO Collection | Double knockout was generated by P1 transduction |
| BW25113 Δ*pyrD*::tGFP-chl$^r$-barcode Δ*glnG*::kan$^r$ | Barcoded library, KEIO Collection | Double knockout was generated by P1 transduction |
| BW25113 Δ*pyrD*::tGFP-chl$^r$-barcode Δ*ihfB*::kan$^r$ | Barcoded library, KEIO Collection | Double knockout was generated by P1 transduction |
| BW25113 Δ*pyrD*::tGFP-chl$^r$-barcode Δ*nupC*::kan$^r$ | Barcoded library, KEIO Collection | Double knockout was generated by P1 transduction |
| BW25113 Δ*pyrD*::tGFP-chl$^r$-barcode Δ*ompR*::kan$^r$ | Barcoded library, KEIO Collection | Double knockout was generated by P1 transduction |
| BW25113 Δ*pyrD*::tGFP-chl$^r$-barcode Δ*phoR*::kan$^r$ | Barcoded library, KEIO Collection | Double knockout was generated by P1 transduction |
| BW25113 Δ*pyrD*::tGFP-chl$^r$-barcode Δ*rfaG*::kan$^r$ | Barcoded library, KEIO Collection | Double knockout was generated by P1 transduction |
| BW25113 Δ*pyrD*::tGFP-chl$^r$-barcode Δ*ubiF*::kan$^r$ | Barcoded library, KEIO Collection | Double knockout was generated by P1 transduction |
| BW25113 Δ*pyrD*::tGFP-chl$^r$-barcode Δ*yfjG*::kan$^r$ | Barcoded library, KEIO Collection | Double knockout was generated by P1 transduction |
| BW25113 Δ*pyrD*::tGFP-chl$^r$-barcode Δ*yohK*::kan$^r$ | Barcoded library, KEIO Collection | Double knockout was generated by P1 transduction |

frame of a single gene was replaced in-frame with a fragment containing turbo GFP, chloramphenicol resistance cassette, and a unique 20 bp sequence that serves as an identification barcode. Since the barcode is the only variable region across strains, it can be amplified from a mixed culture of strains with a single pair of primers. We used primers that amplify a 325 bp region.

## Measurement of bacterial gemcitabine dose–responses and IC50

On the day of the experiment, a 384-well plate containing serially diluted gemcitabine in M9 was prepared at 2× concentration in a volume of 35 µL. For the day-to-day IC50 measurements (*Figure 3B*), a sample from daily evolving populations was directly diluted 1:200 into the microtiter plate with gemcitabine. For measurement of gemcitabine resistance of evolved populations (*Figure 3C*), a sample from frozen last day glycerol stocks of evolution experiment was inoculated into 3 mL LB for overnight culture. Overnight cultures were diluted to OD (600 nm) of 1 in M9 and were added to

the 384-well plate (1:200 final dilution). For single-colony gemcitabine IC50 experiments (*Figure 1B* and *Figure 3D*), single colonies were grown overnight in 3 mL LB and the same dilution protocol was followed as evolved populations. The prepared microplates (with bacteria and gemcitabine dilutions) were incubated at 37°C and 360 rpm double orbital shaking in an automated plate reader (BioTek Eon) and absorbance (600 nm) was monitored every 10 min for 18 hr. All measurements were performed in technical triplicates. Each evolved population is considered as a separate biological replicate (three biological replicates per evolution condition; one biological replicate is shown in the figure for control evolved population). A MATLAB script was used to fit the dose–response curves and infer the IC50 values. Individual growth curves were assessed for quality control and to determine the exclusion criteria for analysis.

## Pooled genetic screen

We thawed a frozen glycerol stock of the pooled barcoded strain collection and inoculated 15 µL of the stock into 25 mL of M9 supplemented with chloramphenicol for overnight growth at 37°C and 200 rpm shaking. In the morning, the culture was diluted to OD (600 nm) of 1, and then diluted 1:400 into 7 ml of M9 with or without 140 µM gemcitabine. We prepared triplicates for each of the two conditions (quadruplicates in validation screens). The tubes were incubated at 37°C shaker and OD was monitored periodically. Once culture crossed OD (600 nm) 0.6, we collected the cells and extracted genomic DNA with a commercial kit (Zymo Quick DNA miniprep Plus Kit, Cat# D4068). Library preparation was identical to the protocol we previously developed (*Rosener et al., 2020*). Briefly, genomic DNA isolated from endpoint of the genetic screen was quantified using Qubit dsDNA high sensitivity kit (Thermo Fisher, Cat# Q32854). We used 6.25 ng DNA to prepare the DNA library. First, barcoded region was amplified using the following primers and 2× KAPA HiFi Hotstart ReadyMix (Kapa Biosystems, Cat# KK2602), which yielded ~350 bp product. PCR products were purified using AMPure XP beads (Beckman Coulter, Cat# A63881). Nextera XT Index Kit protocol (Illumina, Cat# FC-131-1024) was used to add indices and Illumina sequencing adapters to each PCR sample. Next, products were purified using AMPure XP bead (Beckman Coulter, Cat# A63881) purification protocol. The libraries were then run on a 3% agarose gel, and the product was extracted using NEB Monarch DNA Gel Extraction Kit (NEB, Cat# T1020L). Next, we used Agilent High Sensitivity DNA Kit (Agilent Technologies, Cat# 5067-4626) to evaluate the quality and average size of the libraries. Using Qubit dsDNA high-sensitivity kit, we measured the concentration and calculated the molarity of each library. Libraries were normalized to 4 nM, denatured, and diluted according to Illumina MiniSeq/NextSeq System Denature and Dilute Libraries Guide. After pooling, sequencing was performed using MiniSeq High Output Reagent Kit, 75-cycles (Illumina, Cat# FC-420-1001) or NextSeq 500/550 High output Reagent Kit v 2.5, 75-cycles (Illumina, Cat# 20024906). Raw reads were converted to barcode counts using a MATLAB script, which compared a database of all barcodes to the reads (*Rosener et al., 2020*).

We identified the enriched or depleted hits by comparing the relative frequency of individual barcodes when the pooled library grew in the presence or absence of gemcitabine. For this analysis, we used the barcode counts and identified barcodes with significant changes in their relative frequency with DEBRA (*Akimov et al., 2020*). We discarded barcodes with less than 10 counts. We used 'Wald statistical test' and a cutoff value of 16-fold for enrichment and FDR-adjusted p-value of 0.05. Next, we performed gene set enrichment analysis with GAGE (*Luo et al., 2009*) using KEGG (*Kanehisa et al., 2016*) and GO (*Ashburner et al., 2000*) databases at a FDR-adjusted p-value of 0.1. We used published data on the KEIO strain collection (Supplementary file 3 in *Baba et al., 2006*) to classify slow-growing knockouts. Specifically, we used the optical density measurements made after 24 hr of growth of the strain collection on minimal defined media (MOPS) and defined a cutoff value of 0.11 to discriminate normal and slow growth. We chose this cutoff value by the bimodal distribution of the density measurements in the dataset (this value separated the data into two unimodal histograms with 124 slow-growing strains and 4178 strains with normal growth).

## Rapid gemcitabine breakdown assay

We picked from the KEIO strain collection the top 88 gemcitabine-resistant knockouts that were identified in the genetic screen. Knockout strains that were not found in the KEIO collection were picked from the barcoded knockout collection. The strains were grown in LB media with appropriate

antibiotic at 37°C and 200 rpm shaking. We included six overnight cultures of the wild-type strain (BW25113) as controls. The next day, all strains were diluted to OD (600 nm) of 0.5 into 1 mL PBS with gemcitabine (200 µM) in a 96-deep well plate. The plate was incubated in a shaker at 37°C, 900 rpm orbital shaking. 250 µL of the supernatant was sampled after 15 and 45 min and filtered by spinning down at 5000 × *g* using a 96-well plate 0.22 µm filters (PALL Corporation, Cat# 8119). We repeated this procedure for obtaining conditioned buffer three times on different days as independent biological replicates.

After we obtained the conditioned buffers, we evaluated the amount of residual gemcitabine in left by monitoring the growth of a gemcitabine-sensitive reporter strain (*cdd* knockout). The *cdd* knockout was grown overnight in 3 mL M9 media at 37°C, 200 rpm shaking. The next day, the culture was first diluted to OD (600 nm) 1 and then further diluted 1:500 into M9 media. We aliquoted 150 µL of this culture into a 96-well plate and added 50 µL of conditioned buffer to each well. The plate was incubated at 37°C and 360 rpm double orbital shaking in an automated plate reader (BioTek Eon/TECAN). Absorbance (600 nm) was monitored every 10 min for 7 hr. We used the growth measurements from media supplemented with buffer after 15 min of incubation to identify fast degraders and growth measurements from media supplemented with buffer after 45 min of incubation to identify slow degraders. We used a statistical test to identify fast and slow degraders. For this test, we calculated the area under the growth curve (AUC) after blank subtraction for each replicate and used a one-tailed *t*-test to test whether the conditioned buffer from a knockout strain (three biological replicates) reduced or increased the AUC of the reporter strain compared to the buffer prepared with the wild-type strain (18 replicates). We used an FDR-adjusted p-value of 0.1 as a cutoff for statistical significance.

## GC-MS measurement of gemcitabine and dFdU

We picked the knockout strains directly from frozen glycerol stock of the barcoded knockout collection and grew them overnight in 3 mL M9 media at 37°C, 200 rpm shaking. The next day, cultures were washed in saline (distilled water with 0.9% NaCl) and cultures were diluted to an OD (600 nm) of 0.125 in 1350 µL of saline in a 96-deep well plate. Gemcitabine was added to each well to reach a final concentration of 80 µM, and cultures were incubated in microplate shaker at 900 rpm and 37°C. We sampled 450 µL from the cultures at predetermined time points and filtered the samples using 0.22 µm filters by centrifugation at 5000 × *g* for 5 min. We froze the conditioned supernatants at –20°C until the GC-MS measurements were performed. This experiment was performed as three biological replicates (independent three overnight cultures and independent co-incubations).

For GC-MS measurements, first 200 µL of bacterial culture supernatants (or standard solution) were dried under vacuum. Dried samples were derivatized by adding 20 µL of pyridine and 50 µL of *N*-methyl-*N*-(trimethylsilyl) trifluoroacetamide (Sigma-Aldrich, Cat# M-132) followed by incubation for 3 hr at 37°C. The derivatization reaction was allowed to complete for 5 hr at room temperature. Measurements were performed on an Agilent 7890B single quadrupole mass spectrometer coupled to an Agilent 5977B gas chromatograph with an HP-5MS Ultra Inert capillary column (30 m × 0.25 mm × 0.25 µm). Helium was used as carrier gas at flow rate of 1 mL/min (constant flow). The temperatures were set as follows: inlet at 230°C, the transfer line at 280°C, the MS source at 230°C, and quadrupole at 150°C. 1 µL of sample was injected in a splitless mode. Initial oven temperature was set to 80°C, held for 1 min and then increased to 270°C at a rate of 20°C/min, then further increased to 285 at a rate of 5°C/min. MS parameters were three scans/s with 30–500 m/z range, electron impact ionization energy 70 eV. Analytes were identified based on retention time, one quantifier and two qualifier ions that were manually selected using a reference compound. Gemcitabine was quantified as m/z 241 ion eluted at 13.14 min, 2',2'-difluorodeoxyuridine was quantified as m/z 242 ion eluted at 11.42 min and peak integration and quantification of peak areas were done using MassHunter software (RRID:SCR_015040).

## Spheroid experiments

We plated CT26 mouse colon carcinoma cell-line (RRID:CVCL_7256) on 96-well low attachment plates (Costar, Cat# 7007) as 4000 cells/well to form spheroids. Cells were incubated in RPMI 1640 media (Gibco, Cat# 11875-093) with 2 mM L-glutamine, 5% fetal bovine serum (Gibco, Cat# 26140-079) and 25 mM HEPES Buffer (Corning, Cat# 25-060Cl). The plates were centrifuged at 3000 × *g* for 5 min and

kept in a tissue culture incubator with 37°C with 5% $CO_2$. After 4 days of spheroid growth, we serially diluted bacterial cultures into the spheroid microplate and incubated the co-culture for 4 hr with gemcitabine (1.6-fold serially diluted across the columns). Note that all the tested bacterial mutants in this experiment were on *pyrD* knockout background (a pyrimidine auxotroph) to avoid bacterial proliferation in cell culture media that does not contain any nucleotides. Next, the plate was washed with cell culture media with 50 μg/ml gentamicin three times. To achieve this, we used 96-channel handheld electronic pipette (Integra, Viaflo 96) and made use of gravity force. 100 μL media was aspirated capturing the spheroid from the bottom of the wells. After the spheroids sank to the bottom of the tips, the tips were touched to the surface of a fresh plate containing culture media with 50 μg/mL gentamicin, leaving the spheroids in the new plate. We chose a 4 hr time interval to address two opposing requirements of the co-culture system – mitigate overgrowth of the bacterial cultures (which hinders spheroid growth irrespective of the drug) while still allowing enough incubation time to allow for drug degradation. While removal of bacteria after 4 hr may limit the bacterial impact, such a limitation will only result in underestimation of the bacterial impact (but will have no impact on how we evaluate how strains compare to one another).

After three washes, the 96-well plate was transferred to an S3 imaging platform (Incucyte, Sartorius) which is housed inside a tissue culture incubator. The plate was imaged every 6 hr to track spheroid growth and validate that there was no residual contamination of resistant bacteria (evident by bacterial overgrowth). After 7 days of growth, spheroids were washed once using cell culture media with 50 μg/mL gentamicin to get rid of any dead cell and cellular debris and a final microscopy image was captured. We calculated the area of individual spheroids using the Incucyte software (segmentation sensitivity: 40; minimum area filter: 2000 μm²). A MATLAB script was used to make the fitness landscapes by fitting polynomial equations. The following are the steps followed: (1) normalization by timepoint zero: we divided the last day spheroid area to day zero spheroid area (4 days post cell seeding). (2) Normalization by plate: we subtracted the minimum spheroid area from all spheroid areas and divided that value by second largest spheroid area in the plate minus minimum spheroid area. (3) Fitting 3D surface and calculating EC50 lines: we fitted a mesh surface using normalized spheroid areas (2D) using a four-degree polynomial function ('poly44'). Lastly, we marked the EC50 line by calculating the coordinates of the 3D mesh surface where the values corresponded to a mid-response (value of 0.5). This experiment was performed with high resolution of conditions (12 × 8 conditions), which did not require technical replication.

## Lab evolution experiment

We evolved bacteria in sub-inhibitory doses of gemcitabine using a standard serial transfer protocol (200 μM for BW25113, 750 μM for Nissle 1917, 100 μM for F-18) in a deep 96-well plate. For each strain, a single colony was picked for each individual evolution line and grown overnight in M9 media (three biological replicates per condition). The cultures were normalized to OD (600 nm) 1 and diluted 1:200 to a total volume of 1200 μl M9 (with or without gemcitabine). The 96-well plate was incubated at 37°C, 200 rpm shaking and was diluted 1:200 daily into fresh media for a period of 7 days (~53 generations). Resistance of evolving populations was measured daily by diluting the cultures 1:200 into a 384-well plate (35 μl per well) containing serially diluted gemcitabine (prepared in M9 at 2× concentration at a volume of 35 μL). The microplate was incubated at 37°C and 360 rpm double orbital shaking in automated plate reader (BioTek Eon) and absorbance (600 nm) was monitored every 10 min for 18 hr. All absorbance measurements were performed in technical triplicates. All downstream experiments were performed using the frozen last day populations.

## Whole-genome sequencing and analysis of lab evolution experiment

We isolated single individual colonies from last day of the independently evolved populations by streaking them on LB agar plates. Eight colonies were selected and gemcitabine IC50 levels were determined using microtiter plate-based assay. The gDNA was extracted from selected clones using Zymo Quick-DNA Fungal/Bacterial Miniprep Kit (Cat# 11-321). Ancestor gDNAs from the replicates of each strain were pooled at equal ratio and processed as a single sample. DNA sequencing was performed by Seqcenter (Pittsburg, PA). Seqcenter prepared libraries using Illumina DNA Prep Kit and IDT 10 basepair UDI indices. Sequencing was performed on Illumina NextSeq 2000 device (2 × 151 bp reads). For all samples, demultiplexing, quality control, and adapter trimming were performed

with bcl2fastq (Illumina) and trimgalore (Trim Galore, RRID:SCR_011847). DNA sequencing yielded a median coverage of ~120× per reference genome. We used BreSeq tool to identify and annotate mutations (*Barrick et al., 2014*). Mixed ancestor populations were run in breseq population mode to evaluate all existing mutation variants and all other samples were run as pure clones. The following are the NCBI accession numbers for the reference genomes we used in the analysis: CP009273 for *E. coli* BW25113; CP058217.1 for *E. coli* Nissle 1917; and MLZI01000100.1 for F-18. The BreSeq gdtools SUBTRACT/COMPARE was used to substract mutations that existed in the ancestor population (mutations existing 30% or more were considered) from the independently evolved clones. Then, we inspected BreSeq reports to resolve unassigned junction evidence (we did not evaluate the unassigned missing coverage). *Supplementary file 3* includes the mutations identified in all clones after manual inspection of the BreSeq reports. Only the mutations that exist in the evolved clones but not in the ancestor were visualized on concentric circles shown in *Figure 4A* using circa software (OMGenomics). Since F-18 genome consists of 113 contigs, the genomic locations shown on F-18 circa plot are undetermined. Mutations identified on contigs with low/unusual coverage were ignored (these are usually observed on small contigs that are shorter than 1 kb and frequently arise due to challenges in read mapping).

## Obtaining spontaneous nupC mutants

We cultured 96 cultures of the BW25113 strain from individual colonies overnight in LB media (96 biological replicates). In the morning, 100 µL of each culture were plated with glass beads on M9 agar plates containing 0.5 mM gemcitabine. A day after, a single colony, corresponding to single spontaneous resistant mutant, was isolated from each agar plate. A 1.3 kb region spanning the entire *nupC* coding region was amplified by colony PCR and Sanger sequenced with forward (5′ TCACAGGA CGTCATTATAGTG 3′) and reverse (5′ TGAGAGTAATTCATCGGCAC 3′) primers. We annotated mutations by pairwise alignment of the Sanger sequencing results with the *nupC* coding sequence of the reference genome (*Supplementary file 5*). Short insertions and deletions identified in the alignment were annotated as indels. Point mutations identified in the alignment were annotated as missense or nonsense mutations according to their impact on the coding sequence. Insertion of transposon was inferred by a truncated (local) alignment to a region in the *nupC* coding sequence followed by alignment of the remaining sequence to a known transposon. We note that mutations in the promoter region were not sequenced or annotated by this method due to the position of the primers. However, some mutations in this region may account for some of the spontaneous resistant mutants that were not annotated as having *nupC* mutation (*Figure 4B*).

## Luria–Delbruck fluctuation experiment

For each strain, we inoculated four single colonies into 1 mL M9 medium and grew them at 37°C, 200 rpm shaking for 12 hr. Cultures were diluted to OD (600 nm) of 1 in M9. A $10^{-6}$ dilution of OD1 was used to determine accurate CFU/OD (600 nm) by plating on LB agar plates. A $10^{-4}$ dilution of OD 1 was further diluted 27-fold and transferred into 10 wells of a 96-well plate as 200 µL/well (initial population size [$N_0$]: approximately 730 cells/well). The 96-well plate was incubated at 37°C, 1000 rpm shaking overnight. Next day, OD (600 nm) of two wells from the 96-well plate was measured to estimate the final population size ($N_t$). Next, all cultures were diluted 1:40 into 1 mL of 0.9% saline. We plated 200 µL of each cell suspension on M9 agar plates (plating efficiency [$\varepsilon$], 1:40) containing selective amounts of gemcitabine (0.5 mM for BW25113, Nissle 1917, MG1655; 62.5 µM for F-18) or furazolidone (1 mg/mL for BW25113, Nissle 1917, MG1655; 2 mg/mL for F-18). Plates were incubated at 37°C for 18 hr, and the number of colonies was determined. Then, mutation rates were calculated using the RSalvador package in R (*Zheng, 2017*). First, mutation frequency (m) was calculated using the function newton.LD.plating (and 95% confidence intervals were calculated using conf.LD.plating). Then mutation frequency (m) was divided to $N_t$ to find mutation rate per generation (p). These numbers later normalized to a gene that is 1000 bp long.

## Justification for cell lines

Murine colorectal carcinoma cell line (CT26) used in this article was identified as Mycoplasma free, and the cell identity was confirmed by STR profiling (ATCC). According to testing results STR profile was 98% similar to CT26.CL25(ATCC, #CRL-2639). The manuscript refers to this cell line as CT26. This

cell line is not in the list of commonly misidentified cell lines maintained by the International Cell Line Authentication Committee.

## Materials and data availability statement

*E. coli* barcoded library used in this study was kindly provided by Dr. Hirotada Mori. Gemcitabine/no drug evolved BW25113 and F-18 populations are available upon request. Gemcitabine/no drug evolved *E. coli* Nissle 1917 strain and its ancestors cannot be shared with third parties due to material transfer agreement conditions between University of Massachusetts Chan Medical School, USA, and Ardeypharm, GmBH, Germany, unless there is written permission from Ardeypharm, GmBH, Germany.

Raw sequencing reads from barcoded knockout library screen and whole genomes of evolved bacteria are available on NCBI Sequence Read Archive under the bioprojects PRJNA797841, PRJNA911755, and PRJNA855939. R and MATLAB codes used in this article are deposited to GitHub: https://github.com/ss-91/Manuscript-Matlab-Scripts.git (copy archived at swh:1:rev:c18025b9d6f-c557f22a93794c480483c4e48a62e; *Sayin, 2022a*) and https://github.com/ss-91/E.coli-Barcoded-Library-Analysis.git (copy archived at swh:1:rev:e127c5d97069c8f1e5ace7686b303eea93ac4ca9; *Sayin, 2022b*).

## Acknowledgements

The research reported in this article was supported by NIGMS of the National Institutes of Health under award number R35GM133775 and R01AI170722 to AM, by DK068429 and R35GM122502 to AJMW. We would like to thank Dr Elizabeth Shank, Dr Hyun Youk, and Dr Caryn Navarro for their comments on the manuscript. We thank Silvia Ballivian, Matthew Hall, and Dr. Babak Momeni from Boston College for reviewing the manuscript version that was posted on bioRxiv and sharing their review with us. We thank Dr Hirotada Mori for providing us with the barcoded strain collection.

## Additional information

### Funding

| Funder | Grant reference number | Author |
|---|---|---|
| National Institutes of Health | R35GM133775 | Amir Mitchell |
| National Institutes of Health | R01AI170722 | Amir Mitchell |
| National Institutes of Health | DK068429 | Albertha JM Walhout |
| National Institutes of Health | R35GM122502 | Albertha JM Walhout |

The funders had no role in study design, data collection and interpretation, or the decision to submit the work for publication.

### Author contributions

Serkan Sayin, Data curation, Software, Formal analysis, Validation, Investigation, Visualization, Methodology, Writing - original draft, Writing - review and editing; Brittany Rosener, Data curation, Investigation, Methodology; Carmen G Li, Data curation, Software, Formal analysis, Validation, Investigation, Methodology; Bao Ho, Olga Ponomarova, Data curation, Formal analysis, Investigation, Methodology; Doyle V Ward, Resources; Albertha JM Walhout, Supervision, Funding acquisition, Writing - review and editing; Amir Mitchell, Conceptualization, Resources, Data curation, Software, Formal analysis, Supervision, Funding acquisition, Investigation, Project administration, Writing - review and editing

### Author ORCIDs

Serkan Sayin  http://orcid.org/0000-0001-8776-2240
Brittany Rosener  http://orcid.org/0000-0002-1836-8503
Olga Ponomarova  http://orcid.org/0000-0001-6331-9949

Albertha JM Walhout  http://orcid.org/0000-0001-5587-3608
Amir Mitchell  http://orcid.org/0000-0001-9376-3987

**Decision letter and Author response**
Decision letter https://doi.org/10.7554/eLife.83140.sa1
Author response https://doi.org/10.7554/eLife.83140.sa2

## Additional files

### Supplementary files

• Supplementary file 1. Primary and validation screen results.

• Supplementary file 2. Enriched and depleted pathways in the genetic screen using two databases: KEGG: yellow; GO: green, p-adj<0.1.

• Supplementary file 3. Summary of the mutations detected with the BreSeq tool in evolved strains.

• Supplementary file 4. Mutation rates (per kb) for the *nupC* and *nfsA* genes determined by Luria–Delbrück fluctuation experiments.

• Supplementary file 5. List of mutations and their annotations detected in gemcitabine resistant BW25113 colonies.

• MDAR checklist

### Data availability

Barcode Sequencing and whole genome sequencing data have been deposited in NCBI SRA under the bioproject IDs PRJNA797841,PRJNA911755 and PRJNA855939. Supplementary file 1 includes entire numerical data of the original barcoded genetic screen results and validation screen results. Supplementary file 2 includes the all enriched and depleted pathways found in barcoded genetic screen visualized in Figure 1. Supplementary file 3 includes the numerical data from Luria Delbruck Fluctuation experiment which is visualized in Figure 4—figure supplement 2. Supplementary file 4 includes the detailed annotation of genomic mutations found in all evolved clones shown in concentric circa plots at Figure 4A. Supplementary file 5 includes the list of inferred mutations from sanger sequencing files for the data used in Figure 4B.

The following datasets were generated:

| Author(s) | Year | Dataset title | Dataset URL | Database and Identifier |
|---|---|---|---|---|
| Sayin S et al. | 2022 | *E. coli* barcoded deletion library screen for chemotherapeutic antimetabolite gemcitabine | https://www.ncbi.nlm.nih.gov/bioproject/?term=PRJNA797841 | NCBI BioProject, PRJNA797841 |
| Sayin S et al. | 2022 | *E. coli* barcoded deletion library screen for chemotherapeutic antimetabolite gemcitabine (Repeat) | https://www.ncbi.nlm.nih.gov/bioproject/?term=PRJNA911755 | NCBI BioProject, PRJNA911755 |
| Sayin S et al. | 2022 | Whole genome sequencing of gemcitabine evolved *E. coli* strains | https://www.ncbi.nlm.nih.gov/bioproject/?term=PRJNA855939 | NCBI BioProject, PRJNA855939 |

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
