## [Editor Report]

This fundamental work advances our understanding of how bacteria evolve to resist drugs used for cancer treatment and how this could potentially affect drug efficacy and treatment outcome. The data were collected and analyzed using a solid methodology and can be used as a starting point for functional studies of the interaction between the microbiome and cancer drug treatment. The findings will be of broad interest to microbiologists and organismal biologists interested in the role of microbiomes in drug responses.

---

## [Decision Letter]

**Decision letter after peer review:**

Thank you for submitting your article "Evolved bacterial resistance to the chemotherapy gemcitabine modulates its efficacy" for consideration by *eLife*. Your article has been reviewed by 2 peer reviewers, and the evaluation has been overseen by a Reviewing Editor and Christian Landry as the Senior Editor. The following individuals involved in review of your submission have agreed to reveal their identity: Peter J Turnbaugh (Reviewer #1); Alexander Muir (Reviewer #2).

Essential revisions:

1) For the findings to be fully supported, you would need a secondary screen/validation of the primary screen's results.

2) There are some questions regarding the spheroid system that would be best supported with new data but that minimally need to be answered by text edits and more discussions.

3) Some claims regarding how the models used apply to cancer need to be tampered and the limits of the systems used need to be better detailed.

*Reviewer #1 (Recommendations for the authors):*

Key details are missing from the results and methods, a more detailed discussion of the knockout library is essential to interpret the results.

Recommend adjusting the language for the reporter strain assay for gemcitabine activity, since you're not technically measuring "drug degradation". The term "drug activity" is more accurate, which is more agnostic to the mechanism through which the reporter strain growth is altered.

Title – should indicate that this is all in cell culture, no animal models of efficacy are shown.

Line 30 – the line "one third of resistance mutations increase or decrease bacterial drug breakdown" should be removed given that there is a lot of uncertainty about how many total resistance mutations exist and that the assay didn't look directly at drug breakdown.

Line 110 – This is a key point that could be highlighted more in the abstract and discussion. It's fascinating that mutations in a transporter can mask the impact of genes for drug metabolism, which really complicates how you think about the microbiome's role in pharmacology.

Line 125 – the nature of this collection is really vague. Need to describe briefly how this library was made and what it is (transposons? Clean deletions?). Methods should stand on their own without requiring citations.

Line 143 – 16-fold is way too conservative. 2-3 fold is more standard and arguably unnecessary given the p-value cutoff. While a secondary screen would be ideal, at a minimum these data should be re-analyzed at a less conservative cutoff.

Line 190 – specify how many hours.

Line 226 – need to specify how many strains were validated – just 3?

Figure 2c – change to micro symbol no uM.

Figure 3a – the numbers get really small here, with just 8 pure clones tested and 1 sequenced. Need to make sure to adjust the interpretations accordingly. Given this data it's possible to make statements about what was found, but general trends are impossible to infer. Typo: "comensal".

Figure 4 – it's pretty vague in the methods how the mutations were manually curated. Need to provide more details and include the unfiltered output in a supplemental table. As is, it's unclear if nupC is the full signal or if a lot more is going on in these isolates.

Line 494 – need to add citations to support this claim.

Line 666 – typo: "prymidine".

*Reviewer #2 (Recommendations for the authors):*

– My major issue with this manuscript is that the authors make pretty striking claims extending their largely in vitro work on bacterial resistance to gemcitabine to both: (1) in vivo bacterial resistance and (2) tumor gemcitabine response. For example, the abstract says "The two studies […] showing that bacteria-drug interactions transpire locally and systemically and can influence chemoresistance in the host". The introduction further claims "Our work reveals that bacterial adaptation to the frontline chemotherapy drug gemcitabine can take place rapidly and can ultimately increase the chemosensitivity of the hosting tumor", which the authors follow up by stating "Given that administration of antibiotics can be detrimental to cancer patients, our work is potentially impactful since highlights that tumor colonization by chemotherapy metabolizing bacteria may not hamper anticancer treatment in the long run". However, there are no in vivo experiments presented to show that (1) the nupC mutations identified in vitro are actually selected in gemcitabine treated tumors or (2) nupC mutant bacteria affect tumor levels of gemcitabine, gemcitabine action in PDAC cells in a tumor or PDAC response to gemcitabine. I believe to make these extensive claims, the authors would need to provide such in vivo data, at least using animals models of pancreas cancer. In the absence of such data, I would suggest that the authors revise the abstract, introduction and discussion to moderate these extensions from their work to in vivo tumor biology.

– Regarding point (1), the high ex vivo dose of gemcitabine used in these experiments coupled with the rapid doubling rate in culture may lead to selection of resistance mechanisms that are different than those employed in an environment where perhaps the bacteria grow much more slowly and have access to nucleosides that can compete with gemcitabine (PMID: 30827862).

– Regarding point (2), the bacterial conditioning of gemcitabine -> treatment of CT26 cells is not readily extrapolatable to the in vivo PDAC tumor situation. How does the dose of gemcitabine compare to intratumoral/intravenous concentration? How does the # of bacteria compared to cancer cell # in this experiment relate to the population of these cells in tumors? For these reasons, think it is very important to measure how tumors colonized with gemcitabine sensitive versus resistant bacteria impact intratumoral gemcitabine levels, gemcitabine action in PDAC cells and resulting tumor response.

– It is unclear to me why the authors chose to use CT26 organoids for these studies. Would not PDAC cells or organoids be more relevant?

– How do the doses of gemcitabine selected for screening and evolution compare not just to PDAC cell line IC50s but to intratumoral concentrations of gemcitabine?

– In line 219, the authors state they tested an additional 46 resistant strains. What basis was used for the selection of these additional strains.

– The authors have bacterial isolates with pretty different EC50 responses to gemcitabine. Do these isolates have different expression of the genes identified in the screening that could explain the baseline sensitivities of these isolates?

---

## [Author Response]

Essential Revisions (for the authors):1) For the findings to be fully supported, you would need a secondary screen/validation of the primary screen's results.

We repeated the genetic screen in two formats: We repeated the screen with the same collection of knockout strains and we repeated the screen with independent strain collection (knockout strains that were independently cloned). The results from these two completely independent biological replicates are presented on Figure 1—figure supplement 1D. The results (resistance/sensitivity) from both these validation screens were highly correlated with the original screen. We refer to this comparison in the main text (lines 142-148). We provide all results from these validation screens in supp table 1.

2) There are some questions regarding the spheroid system that would be best supported with new data but that minimally need to be answered by text edits and more discussions.

We provide further clarification about the spheroid system in the main text (lines 698-704) and in the response to the detailed points raised by the reviewers. We added independent biological replicates of the spheroid experiments in Figure 2—figure supplement 2.

3) Some claims regarding how the models used apply to cancer need to be tampered and the limits of the systems used need to be better detailed.

We revised the language for specific claims and further clarified the limitations of the system (title, abstract lines 34-36, introduction 107-113, discussion line 504).

Reviewer #1 (Recommendations for the authors):Key details are missing from the results and methods, a more detailed discussion of the knockout library is essential to interpret the results.

We added information on the knockout library in the methods section (lines 551-560).

Recommend adjusting the language for the reporter strain assay for gemcitabine activity, since you're not technically measuring "drug degradation". The term "drug activity" is more accurate, which is more agnostic to the mechanism through which the reporter strain growth is altered.

Done (lines 213-215 and 228-229).

Title – should indicate that this is all in cell culture, no animal models of efficacy are shown.

Done

Line 30 – the line "one third of resistance mutations increase or decrease bacterial drug breakdown" should be removed given that there is a lot of uncertainty about how many total resistance mutations exist and that the assay didn't look directly at drug breakdown.

We revised the statement to indicate that one third of “top” resistance mutations increase or decrease bacterial drug breakdown. This correction makes it clear that our statement holds only for the top hits we tested and not to all resistance mutations.

We further changed the statement to “We infer” (instead of we found) to clarify that the following statement is an interpretation of the results. We think that changes in drug breakdown are a very reasonable interpretation of the functional assay results and we stand behind the use of this term in the abstract.

Line 110 – This is a key point that could be highlighted more in the abstract and discussion. It's fascinating that mutations in a transporter can mask the impact of genes for drug metabolism, which really complicates how you think about the microbiome's role in pharmacology.Line 125 – the nature of this collection is really vague. Need to describe briefly how this library was made and what it is (transposons? Clean deletions?). Methods should stand on their own without requiring citations.

We provided more detailed information about the barcoded library in methods section (lines 551-560).

Line 143 – 16-fold is way too conservative. 2-3 fold is more standard and arguably unnecessary given the p-value cutoff. While a secondary screen would be ideal, at a minimum these data should be re-analyzed at a less conservative cutoff.

See point #2 above.

Line 190 – specify how many hours.

We think it will be confusing to include this information in the figure caption.

Line 226 – need to specify how many strains were validated – just 3?

We next decided to validate the conclusions of our functional assay using an independent chemical approach for the wild-type and three knockout strains. We focused on the fastest and the slowest degraders and cdd knockout as a control (lines 234-237).

Figure 2c – change to micro symbol no uM.

Corrected

Figure 3a – the numbers get really small here, with just 8 pure clones tested and 1 sequenced. Need to make sure to adjust the interpretations accordingly. Given this data it's possible to make statements about what was found, but general trends are impossible to infer. Typo: "comensal".

The typo was corrected.

We observed reoccurrence of nupC inactivation across 12 individual evolution experiments (4 independent replicates in each of the 3 strains). We revised the closing statement and clearly limit the conclusion to what was found (lines 388-390).

Figure 4 – it's pretty vague in the methods how the mutations were manually curated. Need to provide more details and include the unfiltered output in a supplemental table. As is, it's unclear if nupC is the full signal or if a lot more is going on in these isolates.

We revise the methods section (lines 773-779). We provide a new Supplementary file with the raw data (Supplementary file 5).

Line 494 – need to add citations to support this claim.

This sentence refers to a potential route of adaptation and is not supported by published work. We add the word “potentially” to further clarify this statement (line 504).

Line 666 – typo: "prymidine".

Corrected

Reviewer #2 (Recommendations for the authors):– My major issue with this manuscript is that the authors make pretty striking claims extending their largely in vitro work on bacterial resistance to gemcitabine to both: (1) in vivo bacterial resistance and (2) tumor gemcitabine response. For example, the abstract says "The two studies […] showing that bacteria-drug interactions transpire locally and systemically and can influence chemoresistance in the host". The introduction further claims "Our work reveals that bacterial adaptation to the frontline chemotherapy drug gemcitabine can take place rapidly and can ultimately increase the chemosensitivity of the hosting tumor", which the authors follow up by stating "Given that administration of antibiotics can be detrimental to cancer patients, our work is potentially impactful since highlights that tumor colonization by chemotherapy metabolizing bacteria may not hamper anticancer treatment in the long run". However, there are no in vivo experiments presented to show that (1) the nupC mutations identified in vitro are actually selected in gemcitabine treated tumors or (2) nupC mutant bacteria affect tumor levels of gemcitabine, gemcitabine action in PDAC cells in a tumor or PDAC response to gemcitabine. I believe to make these extensive claims, the authors would need to provide such in vivo data, at least using animals models of pancreas cancer. In the absence of such data, I would suggest that the authors revise the abstract, introduction and discussion to moderate these extensions from their work to in vivo tumor biology.

We the removed the last sentence of the abstract to remove any mention of a treated host (lines 34-36). We revised the introduction to clearly indicate that the presented work is only in-vitro and that conclusion regarding antibiotics should only be made if a similar adaptation is observed in treated patients (lines 107-113).

– Regarding point (1), the high ex vivo dose of gemcitabine used in these experiments coupled with the rapid doubling rate in culture may lead to selection of resistance mechanisms that are different than those employed in an environment where perhaps the bacteria grow much more slowly and have access to nucleosides that can compete with gemcitabine (PMID: 30827862).

We revised the text in multiple places (including the title and abstract) to further highlight that this study was performed in-vitro. We agree with the reviewer that growth of bacteria and nutrient availability may be different in the tumor microenvironment. However, very similar claims can be made for the cancer cells themselves (they too likely grow more slowly in the TME and have access to nucleosides). We think that the observation that gemcitabine toxicity in-vitro is comparable in bacteria and in cancer cells is key. We therefore think the premise that within the TME they both likely experience gemcitabine toxicity is a reasonable one.

– Regarding point (2), the bacterial conditioning of gemcitabine -> treatment of CT26 cells is not readily extrapolatable to the in vivo PDAC tumor situation. How does the dose of gemcitabine compare to intratumoral/intravenous concentration? How does the # of bacteria compared to cancer cell # in this experiment relate to the population of these cells in tumors? For these reasons, think it is very important to measure how tumors colonized with gemcitabine sensitive versus resistant bacteria impact intratumoral gemcitabine levels, gemcitabine action in PDAC cells and resulting tumor response.

Measurements of gemcitabine peaks in plasma show that they vary considerably based on the dosing protocol in patients (PMID: 27007129). In extreme cases the plasma concentration can reach 320-512 µM in some patients. We agree with the reviewer that measurement of tumor sensitivity to gemcitabine as a function of the bacterial infection (wt vs. del-nupC) is important but it is clearly outside the scope of this manuscript.

– It is unclear to me why the authors chose to use CT26 organoids for these studies. Would not PDAC cells or organoids be more relevant?

Similarly, to many other chemotherapy drugs, the toxicity mechanism of gemcitabine is quite general and toxicity is expected in numerous cancerous and non-cancerous mammalian cells. We decided to use CT26 cells since this cell-line was used in the original work that described gemcitabine degradation by gammaproteobacteria in pancreatic tumors (PMID: 28912244). This work used CT-26 cells for both in-vivo and in-vitro experiments. CT-26 cells are also routinely used as a workhorse for in-vivo models of infected tumors (in the original work and in other studies). Therefore, the use of this CT-26 cells allows to compare our results to previous relevant studies and to build on our study for future in-vivo work.

– How do the doses of gemcitabine selected for screening and evolution compare not just to PDAC cell line IC50s but to intratumoral concentrations of gemcitabine?

We do not have a reference for intratumoral gemcitabine concentrations. Measurements of gemcitabine peaks in plasma show it can reach a very high concentration (320-512 µM in some patients PMID: 27007129).

– In line 219, the authors state they tested an additional 46 resistant strains. What basis was used for the selection of these additional strains.

We chose the 88 most resistance strains found in our screen. These included the top 42 strains shown in Figure 1E and additional 46 strains. We now clarify this in the text (lines 228-229).

– The authors have bacterial isolates with pretty different EC50 responses to gemcitabine. Do these isolates have different expression of the genes identified in the screening that could explain the baseline sensitivities of these isolates?

We didn’t explore the molecular underpinning for these inter-strain differences. We agree with the reviewer that gene expression level provides a compelling underlying mechanism (however we have preliminary indications that the reasoning is more complex).